# Genomic Epidemiology Unveil the Omicron Transmission Dynamics in Rome, Italy

**DOI:** 10.3390/pathogens11091011

**Published:** 2022-09-04

**Authors:** Maria Francesconi, Marta Giovanetti, Lucia De Florio, Marta Fogolari, Roberta Veralli, Cecilia De Flora, Silvia Spoto, Antonello Maruotti, Elisabetta Riva, Silvia Angeletti, Massimo Ciccozzi

**Affiliations:** 1Clinical Laboratory Unit, Fondazione Policlinico Universitario Campus Bio-Medico, 00128 Rome, Italy; 2Unit of Clinical Laboratory Science, University Campus Bio-Medico of Rome, 00128 Rome, Italy; 3Laboratório de Flavivírus, Instituto Oswaldo Cruz, Fundação Oswaldo Cruz, Rio de Janeiro 22250-060, Brazil; 4Department of Science and Technology for Humans and the Environment, University of Campus Bio-Medico di Rome, 00128 Rome, Italy; 5Unit of Virology, University Campus Bio-Medico of Rome, 00128 Rome, Italy; 6Diagnostic and Therapeutic Medicine Department, University Campus Bio-Medico of Rome, 00128 Rome, Italy; 7Dipartimento di Scienze Economiche, Politiche e delle Lingue Moderne-Libera Università Maria Ss Assunta, 00193 Rome, Italy; 8Medical Statistic and Molecular Epidemiology Unit, University Campus Bio-Medico of Rome, 00128 Rome, Italy

**Keywords:** genomic surveillance, Italy, SARS-CoV-2, Omicron sub-lineages

## Abstract

Since 2020, the COVID-19 pandemic represented an important worldwide burden. Well-structured surveillance by reliable and timely genomic data collection is crucial. In this study, a genomic monitoring analysis of all SARS-CoV-2 positive samples retrieved at the Fondazione Policlinico Universitario Campus Bio-Medico, in Rome, Italy, between December 2021 and June 2022, was performed. Two hundred and seventy-four SARS-CoV-2-positive samples were submitted to viral genomic sequencing by Illumina MiSeqII. Consensus sequences were generated by de novo assembling using the iVar tool and deposited on the GISAID database. Lineage assignment was performed using the Pangolin lineage classification. Sequences were aligned using ViralMSA and maximum-likelihood phylogenetic analysis was performed by IQ-TREE2. TreeTime tool was used to obtain dated trees. Our genomic monitoring revealed that starting from December 2021, all Omicron sub-lineages (BA.1, BA.2, BA.3, BA.4, and BA.5) were circulating, although BA.1 was still the one with the highest prevalence thought time in this early period. Phylogeny revealed that Omicron isolates were scattered throughout the trees, suggesting multiple independent viral introductions following national and international human mobility. This data represents a sort of thermometer of what happened from July 2021 to June 2022 in Italy. Genomic monitoring of the circulating variants should be encouraged considering that SARS-CoV-2 variants or sub-variants emerged stochastically and unexpectedly.

## 1. Introduction

The global COVID-19 pandemic, which started in 2020, had important worldwide repercussions [1]. In particular, it highlighted the need for well-functioning genomic surveillance systems to detect and monitor the evolution of the pandemic on a national and global scale. Reliable and timely genomic data are indeed critical to informing the most appropriate interventions to contain infection [2]. For this reason, the establishment of a well-structured surveillance system appears to be crucial and can be used as a public health monitoring tool [3,4,5,6]. 

Considering the current concern regarding the enhanced prevalence and distribution of different Omicron sub-lineages in Rome, Italy, we carried out a genomic monitoring analysis to understand the dispersion dynamics of the Omicron sub-lineages by analyzing all SARS-CoV-2 positive samples retrieved by the Fondazione Policlinico Universitario Campus Bio-Medico, Rome, Italy between December 2021 and June 2022. 

## 2. Material and Methods

Between December 2021 and June 2022, the Laboratory of the Fondazione Policlinico Universitario sequenced a total of 274 SARS-CoV-2 complete genome sequences belonging to Omicron sub-lineages retrieved from samples that tested positive by real-time (RT) polymerase chain reaction (PCR) for the detection of SARS-CoV-2.

### 2.1. Viral RNA Extraction and RT-PCR Reaction

The Microlab Nimbus (Hamilton) batch loading system was used for SARS-CoV-2 molecular screening by first performing an automatic RNA extraction using both the QiAmp Viral RNA Mini Kit (Qiagen) and the automatic ALtoStar Automation System AM16 (Altona) and then preparing RT-qPCR using the theAllplex™ 2019-nCoV Assay (Seegene).

### 2.2. Whole Genome Sequencing 

A total of 274 clinical samples positive for SARS-CoV-2 infection were submitted to viral genomic amplification and sequencing using the Illumina MiSeqII system, according to the manufacturer’s instructions. Then, consensus sequences were generated by de novo assembling using the iVar with the default setting [7]. All newly generated strains passed the quality control on SOPHIA ddm software and were deposited on the GISAID database (https://www.gisaid.org/, accessed on 27 November 2021). GISAID accession IDs are provided in Appendix A.

### 2.3. Phylogenetic Analysis

Lineage assignment was performed using the Pangolin lineage classification software tool [8] (version 4.0) and revealed that newly SARS-CoV-2 genome sequences belonged to all the BA sub-lineages (BA.1*, n = 124; BA.2*, n = 105; BA.3*, n = 2; BA.4*, n = 18, BA.5*, n = 25). For this purpose, five different datasets were built. Dataset 1 includes a diverse pool of genome sequences (n = 1236) belonging to the BA.1* sub-lineages sampled worldwide and collected up to July 2022, in addition to the 124 genome sequences obtained in this study. Dataset 2 includes a total of 1172 complete genome sequences belonging to the Omicron BA.2* sub-lineage sampled worldwide and collected up to July 2022, in addition to 105 new strains obtained in this study. Dataset 3 includes a total of 736 complete genome sequences belonging to the Omicron BA.3* sub-lineage sampled worldwide and collected up to July 2022, in addition to 2 new strains obtained in this study. Dataset 4 includes a total of 1440 complete genome sequences belonging to the Omicron BA.4* sub-lineage sampled worldwide and collected up to July 2022, in addition to 18 new strains obtained in this study. Dataset 5 includes a total of 1588 complete genome sequences belonging to the Omicron BA.5* sub-lineage sampled worldwide and collected up to July 2022, in addition to 25 new strains obtained in this study. All sequences were aligned using the ViralMSA tool [9] and IQ-TREE2 [10] was used for phylogenetic analysis using the maximum likelihood approach employing the HKY + Γ _nucleotide substitution model, and the SH-aLTR test for statistical robustness. TreeTime [11] was used to transform this ML tree topology into dated trees, after the exclusion of outlier sequences. 

## 3. Results

Between December 2021 and June 2022, the Campus Bio-Medico Hospital, Rome, Italy, participated in regional and national surveys, sequencing complete genomes of the SARS-CoV-2 Omicron variant, and uploading them to the GISAID platform. This could be retrospectively considered a period of transition, since July 2021, the Delta variant was replaced by the new Omicron Variant of Concern and its sub-variants including BA.1 BA.2, BA.3, BA.4, and 30 BA.5, currently diffused worldwide. Several samples were collected and sequenced monthly within this period. The surveillance showed that Omicron was initially detected in December 2021 (5 cases). Subsequently (until June 2022), we were able to sequence and deposit a total of 274 complete genomes of the SARS-CoV-2 Omicron variant, consisting of 125 BA.1*, 105 BA.2, 2 BA.3, 19 BA.4, and 26 BA.5 (Appendix A). The starting samples were all nasopharyngeal or salivary swabs coming from the drive-in in our hospital. All the positive results were shared with the COROnet network (composed of specialized laboratories set up by the Region for laboratory diagnosis of SARS-CoV-2).

Inclusion criteria used for the selection of samples enrolled in the Surveillance for the monitoring of variants in Italy were cycle threshold (Ct) of the PCR lower than the 25th cycle; first positive sample per patient; only one case per household. All the samples sequenced were selected and identified exclusively according to these criteria.

Of the 274 genomes sequenced and published on the GISAID platform in the period between December 2021 and June 2022, 128 were collected from males (46%) and 146 from females (54%). The mean age of the subjects was 58 ± 24 years, and arbitrarily distributed in 4 age groups, namely: 0–20 years, representing 9% of the total, 21–40 years, representing 15% of the total, 41–60 years, representing 28% of the total, and finally, 61–100 years, representing 48% of the total. Throughout Italy, the prevalence of cases due to Omicron variant infection has been equally observed in all age groups, but in our hospital, we observed a higher percentage of samples from infected subjects between 61–100 years.

Our genomic monitoring indicates that all the Omicron (BA.1, BA.2, BA.3, BA.4, and BA.5) sub-lineages were circulating, though BA.1 still was the one with the higher prevalence thought time in this first period (Figure 1A).

More in detail, the first occurrence of the Omicron variant in Italy was detected in a sample collected on 27 November 2021, while the first case of the Omicron variant among the sequencing samples from our hospital was identified on 21 December 2021. Later on, a rapid spread of the variant was observed, and the Omicron variant was later divided into sub-lineages named BA.1 (the main clade), BA.2 and BA.3, BA.4, and BA.5. In June, Omicron was the only variant of SARS-CoV-2 identified in Italy, and at the same time, we were able to notice the increasingly growing trend of BA.5 and a decreasing circulation of BA.2 (https://www.epicentro.iss.it/en/, accessed on 27 November 2021).

Our time-stamped phylogeny revealed that the new Omicron isolates (belonged to the sub-lineages BA.1*–BA.5*) identified in this study are scattered throughout the trees, suggesting that multiple independent viral introductions have occurred through time (Figure 1B–F). Together, our results indicate that virus migration generally followed patterns of national and international human mobility, highlighting how the easing of restriction measures facilitated the spread of the new emerging variants within the country and around the world.

## 4. Discussion

The Omicron SARS-CoV-2 VOC, which was first detected in South Africa in late 2021, has since spread worldwide and has evolved into several sub-variants (BA.1–BA.5). Our genomic surveillance showed the usefulness and peculiarity of phylogenetic tools to understand the evolution over time of the different variants and sub-variants of the Omicron family in Rome, Italy. Our results clearly revealed that the new Omicron isolates (belonged to the BA.1*–BA.5* sub-lineages) identified in this study were scattered throughout the trees, suggesting that multiple independent viral introductions have occurred through time. Genomic surveillance in pandemic events can help also in understanding specific mechanisms of the virus variants in meeting the endemization of SARS-CoV-2. Furthermore, the genomic study of mutations, in certain positions of the genome, can give important information to make predictions about the contagiousness and immunological escape of the different variants. From the epidemiological point of view, the last two variants, Omicron V4/5, seem to have a clear advantage in transmission also with respect to the other more recent sub-variants such as BA.2*. In terms of evolution, this ability is not surprising, because the binding with the hACE2 receptor seems to remain robust [12] despite different and numerous mutations in the spike protein. Obviously, this speculation on phylogenetic and genomic analysis has to be assessed by conducting studies on these variants and sub-variants in animal models [13]. As we can see by the evolutionary analysis, the Omicron lineage has evolved over the past six months of our surveillance and the successive variants and sub-variants seem to become more transmissible as if the evolution gives a major immunological escape. 

## 5. Conclusions

Our genomic surveillance could reflect what happened from December to June 2022 in Italy. We trust that scientific attention remains intently focused on each possible new sub-variant of the Omicron “family”. We have to think that each possible globally dominant variant or sub-variant of SARS-CoV-2 emerged stochastically and unexpectedly, so then genomic monitoring of the circulating variants should be encouraged and extended to a large scale.

## Figures and Tables

**Figure 1 pathogens-11-01011-f001:**
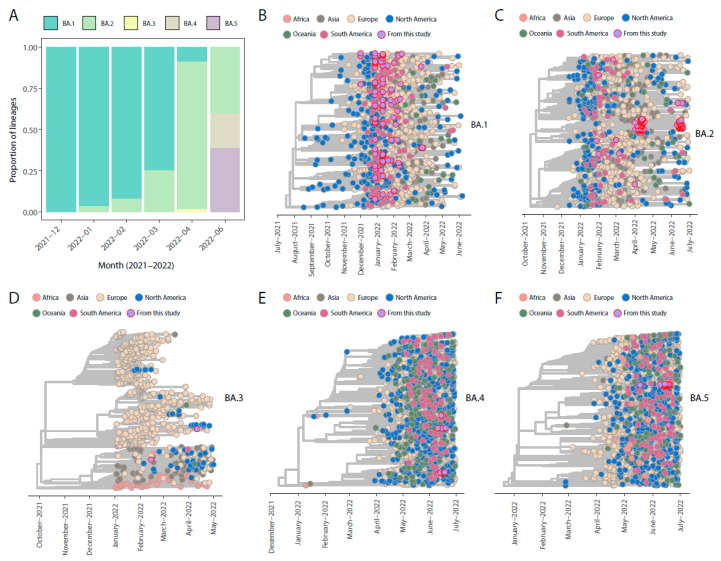
Omicron SARS-CoV-2 dynamic in Italy. (**A**,**B**) Frequency and distribution of SARS-CoV-2 Omicron sub-lineages generated in this study thought time. (**B**–**F**) Phylogenetic characterization of SARS-CoV-2 Omicron sub-lineages genomes sequences generated in this study in addition to reference genomes sampled worldwide. BA.1 tree was reconstructed including n = 1236 BA.1* sub-lineages sampled worldwide, in addition to the 124 genome sequences obtained in this study. BA.2* tree was reconstructed including n = 1172 complete genome sequences sampled worldwide, in addition to 105 new strains obtained in this study. BA.3* tree was reconstructed including a total of n = 736 complete genome sequences sampled worldwide, in addition to 2 new strains obtained in this study. BA.4* tree was reconstructed including 1440 complete genome sequences, in addition to 18 new strains obtained in this study. BA.5* tree was reconstructed including a total of n = 1588 complete genome sequences sampled worldwide, in addition to 25 new strains obtained in this study.

## Data Availability

Data sequences are available on GSAID platform (GSAID.org) and in the Appendix A.

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
