# Peer review of "Genomic Epidemiology Unveil the Omicron Transmission Dynamics in Rome, Italy"

_pathogens, 2022, doi:10.3390/pathogens11091011_

Round 1
Reviewer 1 Report
In the study, authors have sequenced and analyzed 274 SARS-CoV-2 genomes collected in Rome.
There are several issues:
I don't think the collection date of the sample (July 2021-June 2022). I found out that there are 308 complete genomes with low coverage excluded in GISAID during 2021-12-21 and 2022-06-30. There are 4 delta variant samples collected on 2022-01-17. Authors seems to have excluded these delta strain samples (LAZ-UCBM-1374, 1373, 1371, and 1343) intentionally without any explanations or justifications.
Author Response
Reviewer 1:
In the study, authors have sequenced and analyzed 274 SARS-CoV-2 genomes collected in Rome.
Comment 1: There are several issues:
I don't think the collection date of the sample (July 2021-June 2022). I found out that there are 308 complete genomes with low coverage excluded in GISAID during 2021-12-21 and 2022-06-30. There are 4 delta variant samples collected on 2022-01-17. Authors seems to have excluded these delta strain samples (LAZ-UCBM-1374, 1373, 1371, and 1343) intentionally without any explanations or justifications.
Reply: We very much appreciated the reviewer comment. We intentionally excluded in our dataset genomes which presented lower coverage as well as Delta strains, since the main objective was focus on the Omicron variant.
Reviewer 2 Report
The study by Francesconi et al reports the phylogenetic characterisation of 274 sequences obtained in Rome, Italy, between July 2021 and June 2022. The authors use standard phylogenetic methods, including global sequence data to identify the origins and spread of the infections sampled in Rome. The authors show how Omicron was the sole variant in the time period analysed and how Omicron sub-variants changed from BA.1 to BA.2 over time. The authors claim that the phylogeny shows multiple introductions of Omicron variants into Rome.
The study methods are mostly sound, and the article is overall well-written and easy to follow. However, the study lacks in detail and some of the results are not supported by the analysis.
Abstract
There is too much detail for methods and not enough detail on results.
Methods
The authors claim that samples collected between July 2021 and June 2022 where used.
It is not stated how many samples that consists of and how the collection was distributed over time.
The authors stated that 274 samples tested positive for SARS-CoV-2 and all of these classified as Omicron. Later the authors say that the first Omicron positive samples was collected in December 2021. So, what happened with the samples collected between July 2021 and December 2021. Where they all negative for SARS-CoV-2?
More detail on the phylogenetic analysis is needed. What priors were used for IQTree? Bootstrapping? Substitution model?
Results
Overall, more detail is needed.
The study-specific samples are clearly scattered across the phylogeny for BA.1 but the samples seem to form distinct clades for the other sub-lineages. Particularly, BA.2 forms a larger cluster. This should be discussed in more detail. BA.3 shows a geographic specific phylogeny, this should be discussed.
From the global sequences included how many where from Italy? Where do they fall on the phylogeny?
Lines 107 ff. Add numbers to explain the difference in prevalence among the Italian Omicron outbreak and the study samples. Also add a reference.
Figure
The figure is overall well-presented, but it needs more detail and clarifications.
A) Total number of samples included should be added
B-F) the colour used for the study samples are not easy to identify. Maybe using black or white as border colour would be better.
The legend should note the branch length indicates substitution per site per year
Discussion/Conclusion
Line 140. ‘were scattered throughout’ this is not true for BA.2
Line e157. I am not sure what the authors mean by using the term ‘thermometer’
Author Response
Reviewer 2:
General comment: The study by Francesconi et al reports the phylogenetic characterisation of 274 sequences obtained in Rome, Italy, between July 2021 and June 2022. The authors use standard phylogenetic methods, including global sequence data to identify the origins and spread of the infections sampled in Rome. The authors show how Omicron was the sole variant in the time period analysed and how Omicron sub-variants changed from BA.1 to BA.2 over time. The authors claim that the phylogeny shows multiple introductions of Omicron variants into Rome. The study methods are mostly sound, and the article is overall well-written and easy to follow. However, the study lacks in detail and some of the results are not supported by the analysis.
Reply: We thank the reviewer for the positive comment.
Comment 1: Abstract
There is too much detail for methods and not enough detail on results.
Reply: We thank the rewiewer for the comment and we modified abstract.
Comment 2: Methods
The authors claim that samples collected between July 2021 and June 2022 where used.
It is not stated how many samples that consists of and how the collection was distributed over time.
The authors stated that 274 samples tested positive for SARS-CoV-2 and all of these classified as Omicron. Later the authors say that the first Omicron positive samples was collected in December 2021. So, what happened with the samples collected between July 2021 and December 2021. Where they all negative for SARS-CoV-2?
Reply: We appreciate the reviewer comment and apologize for the mistake. The correct collection data is December 2021 and June 2022, we have now provided a supplementary table which in turn include all epidata associated with the strains obtained in this study including the collection date.
Comment 3: More detail on the phylogenetic analysis is needed. What priors were used for IQTree? Bootstrapping? Substitution model?
Reply: Done.
Comment 4: Results
Overall, more detail is needed.
The study-specific samples are clearly scattered across the phylogeny for BA.1 but the samples seem to form distinct clades for the other sub-lineages. Particularly, BA.2 forms a larger cluster. This should be discussed in more detail. BA.3 shows a geographic specific phylogeny, this should be discussed.
Lines 107 ff. Add numbers to explain the difference in prevalence among the Italian Omicron outbreak and the study samples. Also add a reference.
Reply: We appreciate the reviewer comment, but we respectfully disagree with the reviewer since we didn’t detect any monophyletic cluster within the BA* sub-lineages even for the BA.2 (many tips appear to be superimposed in the phylogeny due to the large number of strains). At this stage we will not be able to assume the direction of transmission and the likely-origin of our strains since we should need to have a larger number of strains for the analyzed region. Additionally appear quite important to state that those analysis are absolutely sampling dependent, this mean that our observation might be disrupted by the selection of reference strains and this is why at this stage we didn’t focus to phylogeographically reconstruct the likely-origin of those strains in Rome.
Comment 5: Figure
The figure is overall well-presented, but it needs more detail and clarifications.
- A) Total number of samples included should be added
Reply: We appreciate the reviewer comment. We added this information into the figure legend.
Comment 6:
B-F) the colour used for the study samples are not easy to identify. Maybe using black or white as border colour would be better.
The legend should note the branch length indicates substitution per site per year
Reply: Done.
Comment 7: Discussion/Conclusion
Line 140. ‘were scattered throughout’ this is not true for BA.2
Reply: Changed as required.
Comment 8: Line e157. I am not sure what the authors mean by using the term ‘thermometer’
Reply: We changed the sentence and eliminated the term “thermometer” . the sentence was modified as follows: “Our genomic surveillance could reflect what happened from December to June 2022 in our country”.
Round 2
Reviewer 1 Report
Here are some minor issues which were not resolved.
Line 145: tree should be plural trees since authors built phylogenetic trees in each omicron subvariants.
Line 167: spyke should be spike.
Line 175: 'our country' should be replaced with Italy
Author Response
Reviewer 1:
Comments and Suggestions for Authors
Here are some minor issues which were not resolved.
Line 145: tree should be plural trees since authors built phylogenetic trees in each omicron subvariants.
Line 167: spyke should be spike.
Line 175: 'our country' should be replaced with Italy
Reply: We thank the reviewer for the suggestion, corrections were made following reviewer’s indications
Reviewer 2 Report
none
Author Response
Reviewer 2:
Comments and Suggestions for Authors: none
Reply: We thank the reviewer